# Feasibility, Tolerability and Efficacy of Carfilzomib in Combination with Lenalidomide and Dexamethasone in Relapsed Refractory Myeloma Patients: A Retrospective Real-Life Survey of the Sicilian Myeloma Network

**DOI:** 10.3390/jcm8060877

**Published:** 2019-06-19

**Authors:** Concetta Conticello, Alessandra Romano, Vittorio Del Fabro, Enrica Antonia Martino, Valeria Calafiore, Giuseppe Sapienza, Valerio Leotta, Marina Silvia Parisi, Uros Markovic, Bruno Garibaldi, Salvatore Leotta, Emilia Cotzia, Vanessa Innao, Donato Mannina, Santo Neri, Maurizio Musso, Renato Scalone, Clotilde Cangialosi, Francesco Acquaviva, Giovanni Cardinale, Anxur Merenda, Cinzia Maugeri, Giuseppina Uccello, Massimo Poidomani, Giuseppe Longo, Melania Carlisi, Daniele Tibullo, Francesco Di Raimondo

**Affiliations:** 1Division of Hematology, Azienda Policlinico-OVE, University of Catania, 95125 Catania, Italy; sandrina.romano@gmail.com (A.R.); vdelfabro@yahoo.it (V.D.F.); enricaantoniamartino@libero.it (E.A.M.); valeriacalaf@gmail.com (V.C.); sapienzagius@gmail.com (G.S.); valerio_leotta@yahoo.it (V.L.); marinaparisi@hotmail.it (M.S.P.); urosmarkovic09041989@gmail.com (U.M.); brunga93@gmail.com (B.G.); leotta3@yahoo.it (S.L.); 2U.O.C. Ematologia, ASP8, 96100 Siracusa, Italy; cotemi@hotmail.it; 3Division of Hematology, Dipartimento di Patologia Umana dell’Adulto e dell’Età Evolutiva, Policlinico G. Martino, University of Messina, 98122 Messina, Italy; vinnao@unime.it; 4U.O.C. Ematologia, Azienda Ospedaliera Papardo, 98158 Messina, Italy; donamanni@gmail.com (D.M.); santoneri67@gmail.com (S.N.); 5U.O.C. OncoEmatologia e TMO, Dipartimento Oncologico, La Maddalena, 90146 Palermo, Italy; mamusso53@gmail.com (M.M.); r.scalone69@gmail.com (R.S.); 6U.O.C. Ematologia A. O. Ospedali Riuniti Villa Sofia-Cervello, 90146 Palermo, Italy; clotildecangialosi@gmail.com (C.C.); clotildecangialosi@hotmail.com (F.A.); 7U.O.C. Ematologia, ARNAS-Civico-Di Cristina-Benfratelli, 90147 Palermo, Italy; cardinaleg@libero.it (G.C.); anxur.merenda@arnascivico.it (A.M.); 8Division of Hematology, Sant’Elia Hospital, 93100 Caltanissetta, Italy; maugericinzia@hotmail.com; 9U.O.C. Ematologia, ARNAS Garibaldi, 95122 Catania, Italy; giusy83@virgilio.it; 10SIMT Hematology, ASP7, 97100 Ragusa, Italy; massimo.poidomani@asp.rg.it; 11U.O.C. Ematologia, Ospedale San Vincenzo, 98039 Taormina (ME), Italy; longo_giuseppe@hotmail.com; 12U.O.C. Ematologia, Policlinico P. Giaccone, 90127 Palermo, Italy; melania.carlisi@unipa.it; 13Department of Biomedical and Biotechnological Science, University of Catania, 95125 Catania, Italy

**Keywords:** KRd regimen, multiple myeloma, salvage therapy

## Abstract

**Background:** The ASPIRE (NCT01080391) phase 3 trial showed the efficacy of carfilzomib, lenalidomide and dexamethasone (KRd) triplet for relapse and refractory multiple myeloma (RRMM). However, little is known about safety and efficacy of KRd outside a clinical trial context. **Methods:** Herein we report real life results of KRd given to 130 RRMM patients from 12 Sicilian Centers. **Results:** Median age was 62 years; patients had received a median of two previous lines of treatment (range 1–10) and 52% were refractory to previous treatment. Median number of KRd cycles was 12 (2–29), with a mean duration of treatment of 12 months; 21 patients had received at least 18 cycles. Overall response rate was 61%, including 18% complete response. Median PFS was 22.9 months, median OS was not reached. Creatinine clearance >30 mL/min, quality of the best achieved response and standard Fluorescence In Situ Hybridization (FISH) risk were independent predictors of favorable outcome. Patients who received the full-dosage of carfilzomib in the first two cycles had a better outcome. **Conclusions:** KRd was effective and well tolerated and in a considerable proportion of patients, therapy continued beyond the 18th cycle. The finding of a better outcome in patients with the higher cumulative dose of carfilzomib in the first two cycle encourages to maintain the maximum tolerated dose.

## 1. Introduction

Multiple myeloma (MM) is an incurable hematological cancer due to uncontrolled expansion of neoplastic plasma cells in the bone marrow. Despite the novel insights regarding the biology of the disease and the introduction of very effective second and third-generation novel agents, including lenalidomide, pomalidomide, carfilzomib and ixazomib, the prognosis is still variable based on the high rate of relapsed and refractory patients (RRMM). As a clinical consequence, several lines of treatments are usually required to ensure long-term control of the disease. Moreover, a progressive drug cross-resistance is likely to occur and at each relapse, disease becomes more refractory and the duration of response is progressively shorter [1]. Thus, nowadays one of the major clinical challenge is the choice of the most appropriate salvage therapy for relapsing or refractory patients.

Recent randomized trials have shown the feasibility and efficacy of combination of second and third generation novel agents, at least in the industrialized countries, including the combination of carfilzomib and dexamethasone (ENDEAVOR) [2], panobinostat, bortezomib and dexamethasone (PANORAMA) [3] or daratumumab, bortezomib and dexamethasone (CASTOR) [4]. In other trials the backbone of lenalidomide and dexamethasone has been improved adding carfilzomib (ASPIRE) [5], ixazomib (TOURMALINE) [6], elotuzumab (ELOQUENT-2) [7], or daratumumab (POLLUX) [8]. All these trials have documented an advantage in term of response rate and progression free survival (PFS) in favor of the experimental arm. For example, in the ASPIRE trial, the experimental carfilzomib, lenalidomide and dexamethasone (KRd) cohort had longer median progression free survival (PFS, 26.3 versus 17.6 months) and higher overall response rate than the lenalidomide and dexamethasone (Rd) control cohort, as recently updated. Indeed, median overall survival (OS) was 48.3 months for KRd versus 40.4 months for Rd (hazard ratio, 0.79; *p* = 0.0045) [5,9].

Even if these very encouraging results open new perspectives to ensure deep and durable responses in RRMM, it is still largely unknown how personalize therapy, taking advantage from each line of treatment, and which one could be the most appropriate sequence of treatment for each single patient. Worse yet, several trials have not distinguished relapsed from refractory MM, did not include frail patients (due to e.g., high PS ECOG (Eastern Cooperative Oncology Group Performace Status), impaired kidney function, reduced bone marrow reservoir), and did not distinguish double or single refractory to bortezomib or lenalidomide, making difficult to draw general indications for the clinical practice in real life [10]. Moreover, several drugs are used in different dosages and schedules and we are not still able to define which is the best drug dosage.

Since November 2016 treatment with KRd has been available in Sicily for RRMM. Therefore, in order to evaluate the possibility to translate clinical trial results into efficacy in clinical practice, we retrospectively analyzed a series of Sicilian RRMM patients treated with KRd regimen with the aim to evaluate efficacy and tolerability out of clinical trials. We also investigate the impact on response of previous treatment and of cumulative dose of carfilzomib to provide further insights on the use of this therapy in RRMM patients, whose treatment is currently one of the most important unmet needs [1,10,11,12].

## 2. Methods

### 2.1. Patient Selection

In this real-life retrospective survey, 130 patients were enrolled at 12 Sicilian Centers on behalf of the Sicilian Myeloma Network from November 2016 to December 2018. The study was approved by the independent ethics committee of the coordinating center (n.34/2019/PO) and was conducted in accordance with International Conference on Harmonization guidelines on Good Clinical Practice and the principles of the Declaration of Helsinki. All patients provided written informed consent.

Primary endpoint was the rate of best responses to KRd. Secondary endpoints were time to progression or relapse, progression-free survival, overall survival, and safety.

### 2.2. Procedures and Drug Administration

All patients received intravenous carfilzomib at the dose of 20 mg/m^2^ (30 min) on days 1 and 2, then of 27 mg/m^2^ on days 8, 9, 15 16 of the first cycle and days 1, 2, 8, 9, 15 and 16 of the subsequent cycles, dexamethasone 20 mg on days 1, 2, 8, 9, 15, 16 and lenalidomide 25 mg daily orally on days 1–21 of each 28-day cycle. According to the ASPIRE schedule, treatment with the combination was planned for 18 months followed by treatment with lenalidomide and dexamethasone until progression.

In cases of specific predefined hematological and non-hematological toxic events, carfilzomib and lenalidomide dosages were reduced, according to the manufacturers’ recommendations and medical choice. Treatment was discontinued in cases of disease progression, unacceptable adverse events, or consent withdrawal, while in some centers it was decided to maintain in responding patients the KRd combination beyond the planned 18 cycles, until progression.

### 2.3. Concomitant Medications

One hundred patients received treatment with bisphosphonates every 4 weeks during KRd treatment. An antibiotic and antiviral prophylaxis was carried out with trimethoprim and sulfamethoxazole (800 mg twice a day, twice a week) and acyclovir 200, 400 or 800 mg daily according to the policy of each center. Supportive therapy with erythropoietin (EPO) and granulocyte colony-stimulating factor (G-CSF) was administered accordingly to ASH/ASCO guidelines and policy of each single center [13,14].

### 2.4. Safety and Efficacy Assessment

Each patient’s medical history was recorded on day 1 of each cycle. Physical examinations were conducted, and blood was collected for hematology, renal and liver function tests at each cycle on day 1 and whenever it was considered necessary. Adverse events [15] were graded using the National Cancer Institute Common Terminology Criteria for Adverse Events (NCICTC) criteria (CTCAE) https://evs.nci.nih.gov/ftp1/CTCAE/CTCAE_4.03/CTCAE_4.03_2010-06-14_QuickReference_5x7.pdf.

Efficacy assessment was recorded on day 1 of cycle 2 and every cycle thereafter. Response and progression were reported according to International Myeloma Working Group consensus criteria [16], including complete remission (CR, 100% reduction in M protein according to electrophoresis, with immunofixation negative), very good partial remission (VGPR, ≥90% reduction in serum M protein, and less than 100 mg urine M protein per day), partial remission (PR, ≥50% reduction in serum M protein, and less than 100 mg urine M protein per day), stable disease (SD), progression disease (PD); not valuable (NV). Minimal response was defined according to European Society for Blood and Marrow Transplantation criteria [17].

### 2.5. Statistical Analysis

Descriptive statistics were generated for analysis of results and two-sides *p*-values under 0.05 were considered significant. Qualitative results were summarized in counts and percentages. Overall response rate (ORR) was defined as PR or better (CR + VGPR + PR). OS and PFS were analyzed with Kaplan–Meier tests. Standard errors were calculated by the method of Greenwood, the 95% confidence intervals were computed as 1.96 times the standard error in each direction.

PFS was calculated from the time of inclusion until the date of progression, relapse, death or the date the patient was last known to be in remission. OS was calculated from the time of inclusion until the date of death for any cause or the date the patient was last known to be alive. All calculations were performed using Graph Pad Prism version 5.00 for Windows, Graph Pad Software, San Diego California, CA, USA, www.graphpad.com and MedCalc version 12.30.0.0 (Producer: MedCalc Software bvba, Ostend (Belgium), www.medcalc.org).

## 3. Results

### 3.1. Patients’ Characteristics and Treatment

One hundred and thirty patients were included in the study and evaluated according to an intention-to-treat analysis. Demographic and disease characteristics are shown in Table 1. The series included patients with an ECOG performance status equal to 3 (15%) or 2 (15%), or 0–1 (70%), with a measurable disease defined by International Myeloma Working Group (IMWG) guidelines [16] and have completed at least two cycle of KRd. Data on cytogenetic abnormalities, detected by fluorescence in-situ hybridization on highly purified bone marrow plasma cells, distinguished as high-risk (including t(4;14), t(14;16) and del(17p)) and standard-risk (all the remaining patients), were available in 64 (49%) of patients.

Previous treatments (including number of previous lines of therapy, previous regimen with bortezomib, lenalidomide or pomalidomide, autologous or allogenic transplantation) are shown in Table 2. Half of patients were refractory to previous treatments (52%). Median number of the previous lines of treatment was two (range 1–10).

Of the 130 patients included in the study, at time of analysis, 31 had not yet completed the planned 18 cycles of treatment (18 of them had received at least 12 cycles). Among the 21 patients who completed the 18 cycles, 18 were maintained on treatment (two patients had received 19 cycles, six patients 20 cycles and 10 patients 25 cycles). Seventy-eight patients (60%) discontinued the treatment for the following reasons: progression of disease (63), consent withdraw (four), intensification with high dose chemotherapy and autologous transplantation (four), toxicity (five), secondary malignancy (one), lost to follow up (one).

After a median follow up of 23.9 (range 1–34) months from the start of study treatment, the median number of administered cycles was 12 (2–29) with a mean duration of treatment of 12 months. Fifty-one patients (39%) required drug reduction. Twelve patients shifted to a weekly schedule of carfilzomib and dexamethasone (on days 1–8–15 instead of 1–2, 8–9, 15–16) after two cycles for poor bone marrow reserve and thus completed the planned treatment without any further dose reduction.

### 3.2. Safety

The regimen was well tolerated with grade 3–4 hematological and non-hematological adverse events affecting 46 (35%) and 48 (36%) patients respectively. Grade 3–4 hematological AEs occurred in 35%, anemia in 12%, neutropenia in 15%, thrombocytopenia in 15%. The most common non-hematological AEs were pneumonia (8%) and cardiovascular (CV, 8%, 11 patients: eight cases of hypertension, two cases of heart failure and one case of atrial flutter) (Table 3). In patients who developed serious AE, KRd dose reduction (51, 39%) and discontinuation (two, 0.1%) were applied.

Platelet transfusions support was required in five patients (0.3%). EPO support was used in 48 (37%) patients, and red blood cells transfusion in 12 patients (9%). Secondary prophylaxis with G-CSF administration (filgrastim 30 MU for two days every two weeks) was required to support 20 patients (15%). Thanks to the concomitant antibiotic and antiviral prophylaxis, none of the patients had *Herpes zoster* reactivation or *Pneumocystis jurovecii* related-pneumonia. Nausea and vomiting were mild (grade 1–2), affecting 10% of patients.

In 22 (17%) patients with reduced renal function at baseline, not requiring dialysis, carfilzomib was given at full dosage and lenalidomide 5 mg/day on days 1–21; in 8/22 non-dialyzed patients renal impairment improved progressively during KRd up to normalization of estimated glomerular filtration rate.

### 3.3. Efficacy

Overall response rate was 61% (79 patients); 24 patients (18%) achieved a complete response (CR), 25 patients (19%) achieved a very good partial response (VGPR), 30 (23%) achieved PR, 18 (13%) a minimal response (MR) and nine (7%) had a stable disease. Therefore, a clinical benefit was observed in 101 patients (77%). Progression occurred in 24 patients (22%), including 14 (11%) who did not achieve any response and 10 patients that achieve a minimal response within a median of six cycles (range 3–16) and then progressed. Data on efficacy are shown in Table 4. The median number of cycles to obtain the maximal response was six (range 2–9), in 7/79 it was obtained within the first two cycles.

The quality of achieved response improved cycle by cycle. Among the 21/130 (16%) patients that had received at least 18 cycles, ORR improved to 85% (including 40% CR, 20% VGPR, 25% PR). Among these 18 patients who received more than 18 cycles, three patients exhibited improvement in the quality of achieved response (two PR from MR and one CR from PR) while the others maintained the achieved response (including six patients who maintained CR) (Table 4).

The median progression free survival (PFS) was 22.9 months (Figure 1A), while the median overall survival (OS) was not reached (Figure 1B). The 2-years-PFS was longer in those patients who achieved VGPR or better than in those who did not (66.9 versus 33.3%, *p* = 0.0002, HR 3.19, 95% C.I. 1.53–6.64, Figure 2A). In the group of 64 patients with available FISH, the 19 patients with high-risk abnormalities had shorter median PFS than standard-risk (8.4 months versus unreached, *p* < 0.0001, HR 3.69, 95% C.I. 1.53–8.88, Figure 2B). Similarly, 2-years-OS was longer in those patients who achieved VGPR or better than in those who did not (78.0 versus 51.2% *p* = 0.0008, HR 4.50, 95% C.I. 1.79–11.33, Figure 2D) and in standard risk FISH (unreached versus 11.6 months, *p* = 0.0001, HR 4.59, 95% C.I. 1.64–12.88, Figure 2E).

Median cumulative carfilzomib received dose was 4000 mg/m^2^. Since all patients had received at least two KRd cycles, we evaluated the prognostic significance of carfilzomib cumulative dose limited to the first two cycles and we found that patients who received a cumulative dose of at least 310 mg/m^2^ (equivalent to full-dose) in the first two cycles, had longer median PFS (27.0 vs. 14.5 months, HR 1.99, 95% C.I. 1.21–3.31, *p* = 0.006, Figure 2C). However, median OS was not reached in both groups (Figure 2D), and there was not a statistically significant difference.

### 3.4. Predictors of Outcome of KRd in the Real-Life Setting

In the attempt to identify patients that could obtain the most advantage by KRd treatment, univariate analysis of PFS and OS showed that previous exposure to more than two lines of treatment, including previous exposure to lenalidomide, was associated to shorter median PFS and OS (Table 5, Figure 3). However, it should be noted that patients previously treated with lenalidomide, although with a lower ISS stage, had a higher number of prior treatments (Appendix A). Despite the small number of patients who received pomalidomide before KRd (N = 16), we found that also prior exposure to pomalidomide was a negative predictor of PFS, but not OS, probably for the enrichment of other bad prognostic factors (Table 5, Appendix A).

Both PFS and OS were affected by kidney function, as shown in univariate and multivariate analysis (Table 5, Figure 4). Indeed, median PFS and OS were longer in patients with clearance creatinine >30 mL/min (respectively, 27.0 vs. 8.4 months, *p* = 0.0003, HR 2.72, 95% C.I. 1.24–5.98 and unreached median vs. 10.3 months, *p* < 0.0001, HR 3.59, 95% C.I. 1.47–8.78).

Previous exposure to thalidomide, autologous Bone Marrow Transplantation (BMT), sex, age, type of relapse (biochemical versus aggressive), did not affect neither PFS nor OS of patients treated with KRd. However, supportive treatment to prevent skeletal events with zoledronic acid was weakly associated to longer median OS than who did not receive (unreached vs. 18.2 months, *p* = 0.018, HR 0.49, C.I. 95% 0.24–1.00).

Thus, univariate analysis disclosed that the predictors of longer PFS were the number of prior lines received (<2, *p* = 0.0001), lack of previous exposure to lenalidomide (*p* = 0.0001) and pomalidomide (*p* = 0.02), a cumulative dose of carfilzomib of 310 mg/mq within the first two cycles (*p* = 0.006), creatinine clearance >30 mL/min (*p* = 0.0003) and high-quality response (at least VGPR, *p* = 0.0002, Table 5). In multivariate analysis the number of prior lines received (*p* = 0.04, HR 1.78, 95% C.I. 1.00–3.16), creatinine clearance >30 mL/min (*p* = 0.0002, HR 3.34, 95% C.I. 1.77–6.29) and a high-quality response (*p* = 0.002, HR 3.19, 95% C.I. 1.53–6.64) maintained independent significance as predictors of longer PFS.

The same factors could predict better OS in univariate analysis (Table 5). Multivariate analysis disclosed that concomitant exposure to zoledronic acid (*p* = 0.03, HR 0.49, 95% C.I. 0.26–0.94), creatinine clearance >30 mL/min (*p* = 0.0003, HR 3.89, 95% C.I. 1.87–8.07) and a high-quality response (*p* = 0.0015, HR 4.5, 95% C.I. 1.79–11.33) were independent predictors of OS in patients who received KRd.

Despite the small sample size, in the sub-group of 64 patients with available FISH, high-risk FISH was associated to shorter both PFS and OS, in both univariate and multivariate analysis (respectively, HR: 4.28, C.I. 95% 2.09–8.79, *p* = 0.0001 and HR: 4.18, C.I. 95% 1.92–9.11, *p* = 0.0003, Figure 2, Table 5).

In the 21 patients who received at least 18 cycles, univariate and multivariate analysis confirmed clearance creatinine >30 mL/min as a positive predictor of PFS and OS (Appendix A).

## 4. Discussion

In this real-life survey, we evaluated the feasibility and effectiveness of KRd in RRMM patients of 12 Sicilian Centers, belonging to the Sicilian Myeloma Network, outside of controlled clinical trials.

We confirmed that KRd is safe and feasible, the most common hematological side effect being asymptomatic thrombocytopenia and, among the non-hematological side effects, gastrointestinal, pneumonia and, to a lesser extent, cardiac ones were observed. There are several warnings about increased cardiotoxicity in MM patients treated with proteasome inhibitors but there is an unquestionable clinical benefit of carfilzomib in combination with lenalidomide and dexamethasone on PFS and OS in RRMM patients [9]. Reduction and management of cardiovascular risks has therefore become a priority in the management of these patients [18].

We found a correlation between cumulative carfilzomib dose in the first two cycles and PFS and OS, a parameter that closely reflects the dose-dependent effect of carfilzomib, although it should be underlined that patients who did not receive full dose of carfilzomib had often significant comorbidities, likely contributing to poorer outcome. This observation is consistent with previous reports about improved clinical outcomes associated to increased first-in-class-proteasome inhibitor bortezomib exposure [19]. Our findings further support the hypothesis that higher doses of carfilzomib are associated to improved outcome. The last interim analysis of the randomized, open-label phase III A.R.R.O.W. (NCT02412878) trial, that compared RRMM patients treated with carfilzomib once a week (70 mg/m^2^) or twice a week (27 mg/m^2^), showed that weekly carfilzomib at 70 mg/m^2^ significantly prolonged PFS, while in the control arm PFS and ORR were inferior to the phase III ASPIRE trial [20]. In a meta-analysis of 14 trials, combination regimens and higher carfilzomib dose offered better response with no significant extra toxicity [21]. Our data therefore confirm that in order to ensure the largest and deepest clinical benefit, carfilzomib should be administered without reduction and adverse events should be prevented and faced with a proactive management. To this purpose, supportive care with antibiotic prophylaxis associated to growth factors have a pivotal role. Indeed, in our series only 21 patients interrupted treatment due to infections or hematological toxicities.

Our experience also indicates that it is possible to continue KRd treatment beyond the 18 cycles that were planned in the ASPIRE trial. Actually, among the 21 patients who completed the 18th cycle, 18 were maintained in treatment and three achieved a better response, without adding further toxicity. These results identify a specific population with good sensitivity to KRd therapy beyond 18 cycles; further studies are needed to confirm whether this prolonged exposure to KRd is recommended.

ORR was lower than what previously reported in the ASPIRE trial (61 versus 87%), with a comparable proportion of patients in CR [5]. In addition, the median PFS in our real-life experience was 22.9 months, weakly shorter than the 26 months reported in the ASPIRE trial. It should be underlined that in comparison with the study population of the ASPIRE trial [5], our patients had inferior performance status (15% of patients in our series had PS ECOG higher than three, 15% equal or higher than two while 90% of patients enrolled in ASPIRE had 0–1), received higher cumulative number of previous treatment, (range 1–10 versus 1–3), had an higher percentage of lenalidomide (40 vs. 20%) or even pomalidomide (12 vs. 0) exposure, had myeloma-related-renal injury (17% vs. 0), including some patients on dialysis, and three patients had already received an allogenic stem cell transplantation (Table 6).

Further sub-analysis of the ASPIRE showed that PFS was shorter in those patients who received KRd after previous exposure to lenalidomide [22]. Indeed, in the ASPIRE trial the superiority of KRd over Rd was only 5 months in patients previously treated with lenalidomide and further reduced to two months in lenalidomide refractory patients. Our findings confirm that also in real-life previous exposure to lenalidomide could reduce the KRd efficacy.

The difficulty in treating lenalidomide-refractory patients with carfilzomib is an important finding. In this regard, our data weaken the hypothesis that a patient resistant to lenalidomide can be easily rescued by adding a different class of drugs and should have a good response when combined carfilzomib and dexamethasone. This issue is currently under investigation by several groups. For example, in the ENDEAVOR trial, that compared the association of high-dose carfilzomib and dexamethasone (Kd) versus bortezomib and dexamethasone (Vd) in RRMM patients, lenalidomide refractory patients had dismal outcome, with a median PFS of only 8.6 months in Kd arm, not very different from the median PFS of 6.6 months in the Vd arm [23]. Kd elicited better outcome (PFS 18.7 months) in those patients who received one prior line of treatment containing bortezomib than lenalidomide (PFS 15.6 months) [23]. These observations are topical since several studies have indicated a clinical benefit of a maintenance therapy with lenalidomide in both transplant eligible [24,25] and non-transplant eligible patients [26], either for a fixed duration (18 months) or given continuously as shown by the FIRST trial [27]. Therefore, in the future an increasing number of MM patients will relapse during a treatment with lenalidomide and should be considered as lenalidomide refractory [1,16,22,28].

Finally, in the ASPIRE trial, KRd was able to overcome the main factors involved in the bad outcome of RRMM, but results were less encouraging in the presence of high-risk FISH abnormalities. Despite the small number of patients with an evaluable FISH in our series, we found that high-risk RRMM patients still have a dismal outcome and should be addressed to clinical trials.

## 5. Conclusions

Our findings suggest that the ideal candidate for treatment with KRd should be a patient who relapses after a thalidomide- or bortezomib-based regimen, independently from prior autologous stem cells transplantation, as previously shown by ASPIRE [22] trial. Therefore, in the real life, where many patients receive KRd after being treated with lenalidomide, we shouldn’t expect the same brilliant results of the ASPIRE [5]. However, thanks to its low toxic profile, carfilzomib could be combined with drugs with different mechanisms of action other than lenalidomide. Our previous experience with a combination of bortezomib, doxorubicin, cyclophosphamide, and dexamethasone; or bortezomib, melphalan and dexamethasone [29,30] supports the idea that even patients that have been heavily pre-treated with the novel agents can retain a sensitivity to alkylating agents. Unfortunately, there are limited data on carfilzomib as part of other combinations, especially with alkylating agents, like melphalan. In newly-diagnosed elderly MM patients’ KMP did not improve PFS compared with VMP, suggesting that melphalan may not be an ideal drug to combine with carfilzomib in this setting [31]. Preliminary results of the MyelomaUKfive (MUKfive) trial are expected in the next year to evaluate carfilzomib in combination with cyclophosphamide and dexamethasone in RRMM.

## Figures and Tables

**Figure 1 jcm-08-00877-f001:**
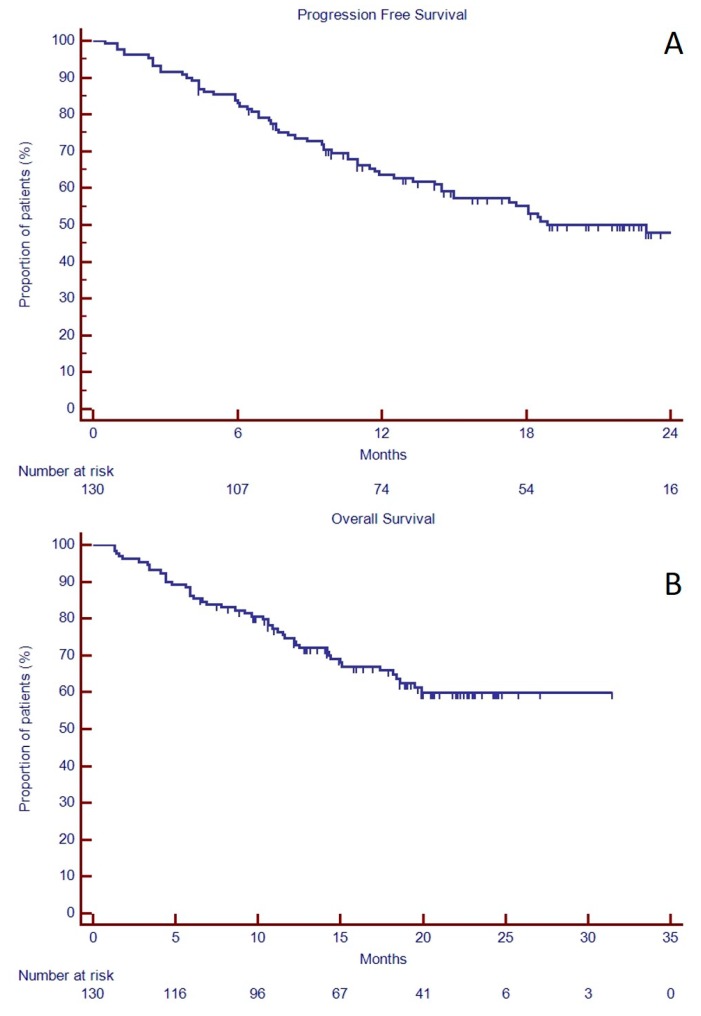
Progression free survival (**A**) and overall survival (**B**) in 130 relapse and refractory multiple myeloma (RRMM) candidates to KRd as salvage regimen in a real-life approach.

**Figure 2 jcm-08-00877-f002:**
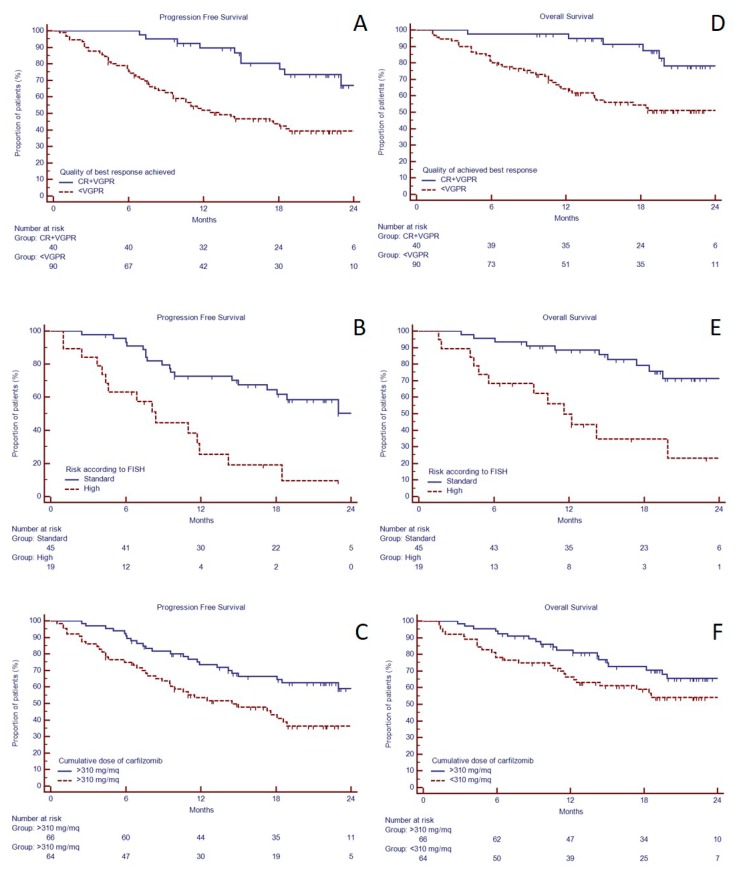
Progression free survival and overall survival in 130 RRMM candidates to KRd as salvage regimen in a real-life approach. Progression free survival in 130 RRMM candidates to KRd as salvage regimen in a real-life approach based on response achieved (**A**), FISH risk (data available for only 64 patients) (**B)**, and cumulative dose of carfilzomib received (**C**). Overall survival in 130 RRMM candidates to KRd as salvage regimen in a real-life approach based on response achieved (**D**), FISH risk (data available for only 64 patients) (**E**), and cumulative dose of carfilzomib received (**F**).

**Figure 3 jcm-08-00877-f003:**
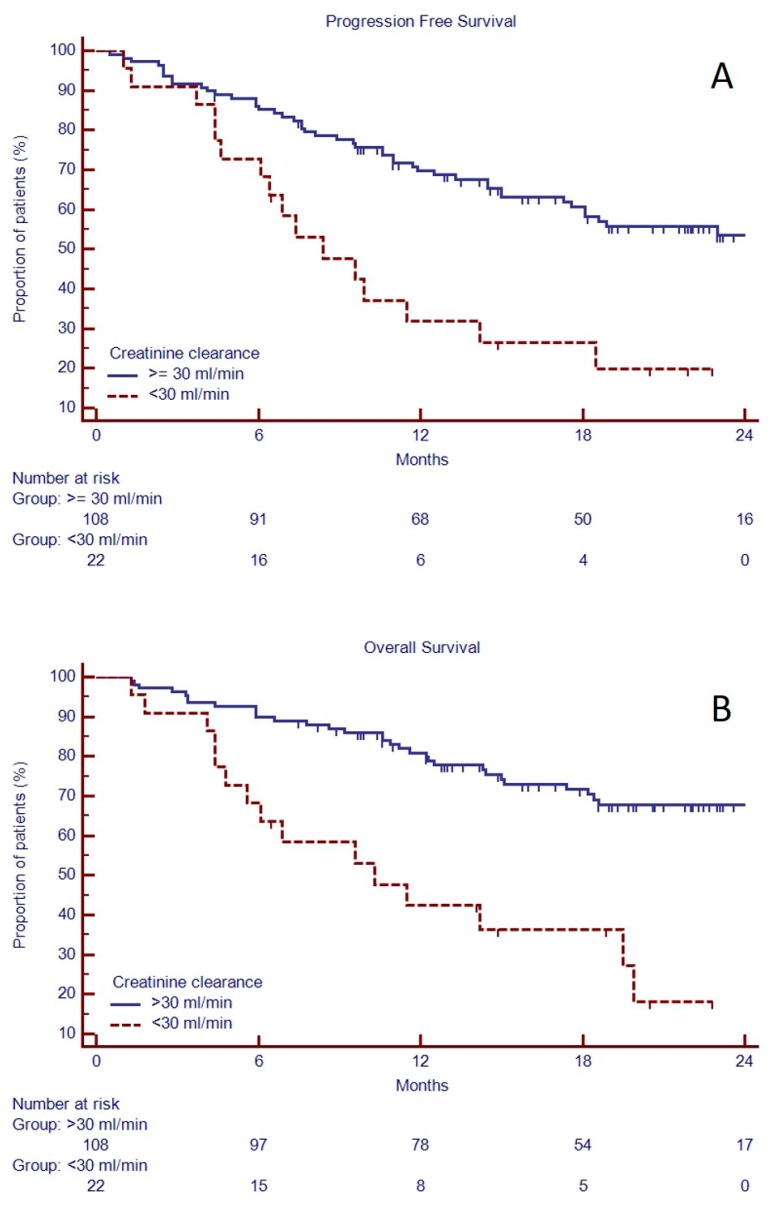
Progression free survival (**A**) and overall survival (**B**) based on previous exposure to lenalidomide in 130 RRMM candidates to KRd as salvage regimen in a real-life approach.

**Figure 4 jcm-08-00877-f004:**
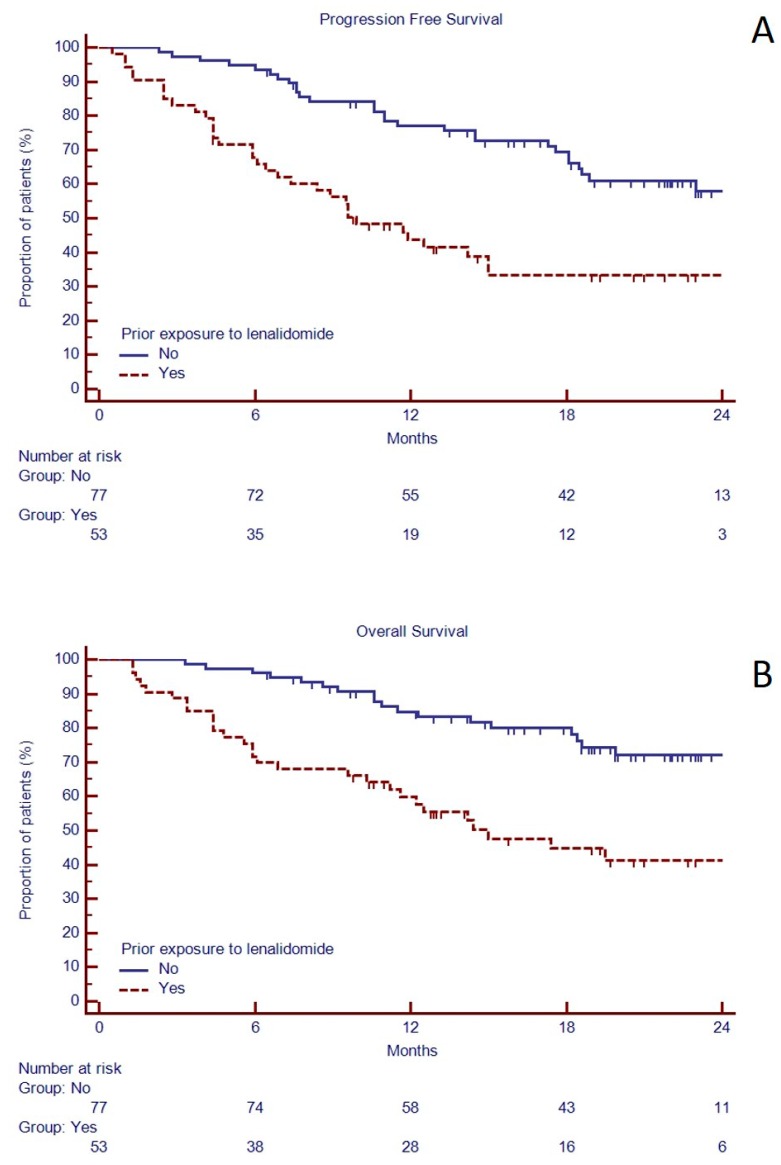
Progression free survival (**A**) and overall survival (**B**) based on creatinine clearance in 130 RRMM candidates to KRd as salvage regimen in a real-life approach.

**Table 1 jcm-08-00877-t001:** Patient characteristics of 130 relapse and refractory multiple myeloma (RRMM) candidates to carfilzomib, lenalidomide and dexamethasone (KRd) as salvage regimen in a real-life approach.

Characteristics	130 pts
**Median age (range)**	62 (33–86)
**Males/Females, *n* (%)**	73/57 (56/44)
**Serum isotype**	
IgG, *n* (%)	77 (60)
IgA, *n* (%)	25 (19)
IgM, *n* (%)	1 (0.5)
Micromolecular, *n* (%)	25 (19)
Non-secretory, *n* (%)	2 (1)
**ISS**	
I, *n* (%)	46 (35)
II, *n* (%)	43 (33)
III, *n* (%)	41 (32)
**Durie and Salmon staging**	
IA, *n* (%)	33 (25)
IIA, *n* (%)	15 (12)
IIB, *n* (%)	4 (3)
IIIA, *n* (%)	56 (43)
IIIB, *n* (%)	10 (8)
Unknown, *n* (%)	12 (9)
**FISH analysis available**	**64 (49)**
Standard Risk, *n* (%)	45 (70)
High risk, *n* (%)	19 (30)
**>3 bone lesions, *n* (%)**	70 (54)
**Clearance creatinine <30 mL/min, *n* (%)** **Dialyzed patients, *n* (%)**	22 (17) 5 (4)
**Extramedullary disease, *n* (%)**	4 (3)

**Table 2 jcm-08-00877-t002:** Prior regimens used of 130 relapse and refractory multiple myeloma (RRMM) candidates to KRd as salvage regimen in a real-life approach.

Previous Regimens	
**Median (range)**	2 (1–10)
1–3 regimens	114 (88)
≥4 regimens	16 (12)
**Previous exposure to novel agents, pts *n* (%)**	
Bortezomib	126 (96)
Lenalidomide	53 (40)
Thalidomide	78 (60)
Pomalidomide	16 (12)
**ASCT (Autologous Stem Cell Transplant)**	
Single	40 (30)
Double	40 (30)
No	50 (38)
**AlloSCT (Allogenic Stem Cell Transplant)**	3 (2)

**Table 3 jcm-08-00877-t003:** Hematological and non-hematological toxicities in real-life KRd.

	Events	
Hematological Toxicity	All Grades	3–4 Grade
Anemia, *n* (%)	46 (35)	15 (12)
Neutropenia, *n* (%)	32 (25)	19 (15)
Piastrinopenia, *n* (%)	39 (30)	19 (15)
**Non Hematological Toxicity**	**All Grades**	**3–4 Grade**
Gastrointestinal, *n* (%)	40 (31)	4 (3)
Pneumonia, *n* (%)	37 (28)	11 (8)
Cardiovascular, *n* (%)	25 (19)	11 (8)
Neurologic, *n* (%)	20 (15)	6 (5)
Fatigue, *n* (%)	15 (12)	2 (2)
Vascular, *n* (%)	9 (7)	6 (5)
Diabetes, *n* (%)	2 (1)	1 (1)
Cutaneous, *n* (%)	8 (6)	1 (1)
Pulmonary, *n* (%)	4 (3)	3 (2)
Ophthalmological, *n* (%)	3 (2)	0 (0)
ORL, *n* (%)	9 (7)	2 (2)
Musculoskeletal, *n* (%)	7 (5)	0 (0)
Hepatic, *n* (%)	4 (3)	2 (2)
Renal, *n* (%)	2 (1)	1 (1)

**Table 4 jcm-08-00877-t004:** Response to KRd in 130 relapse and refractory multiple myeloma (RRMM) treated with KRd as salvage regimen in a real-life approach. according to International Myeloma Working Group (IMWG) criteria.

	Best Response in Patients Who Received Less Than 18 Cycles	Best Response in Patients Who Received at Least 18 Cycles	Best Response of All Patients
	*n* = 109 (100%)	*n* = 21 (100%)	*n* = 130 (100%)
**ORR**	61 (55)	18 (85)	79 (60)
CR	15 (14)	9 (40)	24 (18)
VGPR	21 (19)	4 (20)	25 (19)
PR	25 (22)	5 (25)	30 (23)
**MR**	**16 (15)**	**2 (10)**	**18 (14)**
**SD**	**8 (7)**	**1 (5)**	**9 (7)**
**PD**	**24 (22**)	**0 (0)**	**24 (19)**

**Table 5 jcm-08-00877-t005:** Univariate and multivariate analysis of PFS and OS.

			Univariate Analysis		Multivariate Analysis		Univariate Analysis		Multivariate Analysis	
		*n*	PFS @ 24 Months	*p*-Value	HR (95% CI)	*p*-Value	OS @ 24 Months	*p*-Value	HR (95% CI)	*p*-Value
**Age**	≤65	68	50.5	0.66			66.4	0.12		
	>65	62	42.6				55.4			
**Gender**	male	73	55.1	0.20			55.6	0.98		
	female	57	38.1				62.8			
**ISS**	I–II	95	50.4	0.14			60.8	0.37		
	III	35	43.5				57.8			
**Prior lines**	1–2	95	56.7	***0.0001***	1.78 (1.00–3.16)	***0.04***	70.0	***0.0002***	1.81 (0.93–3.52)	0.08
	more than 2	35	22.4				33.5			
**Previous lenalidomide**	no	77	58.0	***0.0001***	0.79 (0.43–1.46)	0.46	72.2	***0.0001***	1.84 (0.96–3.54)	0.07
	yes	53	33.3				41.4			
**Previous pomalidomide**	no	114	51.2	***0.02***	1.34 (0.60–3.02)	0.46	62.4	0.17		
	yes	16	24.2				42.2			
**Previous BMT**	no	50	33.4	0.29			55.7	0.58		
	yes	80	53.2				61.8			
**Zoledronic acid exposure**	no	31	41.4	0.11			47.6	***0.018***	0.49 (0.26–0.94)	***0.03***
	yes	99	52.6				63.4			
**Relapse**	biochemical	31	59.1	0.54			62.0	0.37		
	clinical	99	47.6				52.6			
**ClCr**	<30 mL/min	22	19.9	***0.0003***	3.34 (1.77–6.29)	***0.0002***	18.2	***<0.0001***	3.89 (1.87–8.07)	***0.0003***
	>30mL/min	108	53.5				67.7			
**K Cumulative dose (2 cycles)**	<310mg/mq	64	36.4	***0.006***	1.40 (0.81–2.46)	0.23	54.2	***0.08***		
	>310mg/mq	66	59.1				65.5			
**Best response**	<VGPR	90	33.3	***0.0002***	3.19 (1.53–6.64)	***0.002***	51.2	***0.0008***	4.50 (1.79–11.33)	***0.0015***
	CR + VGPR	40	66.9				78.0			
**FISH risk ***	Standard	45	50.2	***<0.0001***	4.28 * (2.09–8.79)	***0.0001***	23.0	***0.0001***	4.18 * (1.92–9.11)	***0.0003***
	High	19	9.0				71.3			

* available data for 64 patients only.

**Table 6 jcm-08-00877-t006:** Comparison of patients’ characteristics at baseline and response to KRd in real life and in the ASPIRE clinical trial.

	Real Life Study	Aspire Trial
Performance Status ECOG > 3	15%	0
Previous Lines median (range)	2 (1–10)	2 (1–3)
Creatinine ≤ 30 mL/min	17%	0
Patients on Dialysis	4%	0
Prior Lenalidomide exposure	40%	20%
High risk patients	21%	12%
Prior pomalidomide exposure	12%	0%
Prior ASCT	60%	56%

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
