# Peer review of "Feasibility, Tolerability and Efficacy of Carfilzomib in Combination with Lenalidomide and Dexamethasone in Relapsed Refractory Myeloma Patients: A Retrospective Real-Life Survey of the Sicilian Myeloma Network"

_jcm, 2019, doi:10.3390/jcm8060877_

Reviewer 1 Report

Introduction:

p2 Line 68: I wouldn't call the prognosis of myeloma dismal anymore.  R-ISS 1 patients do not have a median survival.  I would suggest revising to say that relapse is inevitable instead. 

p3 Line 89: Grammar error.  "To worse this scenario" should be "Worse yet, several trials have not distinguished relapsed from refractory MM,..."

Methods:

Why was efficacy assessment delayed until after Cycle 2? The median time to response with KRd is 1 cycle.  Were patients not assessed (i.e. did not reach cycle 2) reported as having not reached PR or progressed or were they censored?  Please clarify in the text.  

Why were efficacy assessments performed every two months?  This would introduce a significant PFS bias, since many PFS events would be recorded a month late.  The ASPIRE study, upon which the study treatment protocol was based on, had efficacy assessments on Day 1 of each cycle. 

p4 line 153: you left the entire reference written out in the text, when a foot note would be more appropriate.

Statistics: please specify if the p-value was 2-sided or not. 

Results:

page 5: safety: 8% of the patients had Grade 3 or higher cardiovascular complications.  CV AEs are of special interest when carfilzomib is used.  Can you please clarify what the CV complications were (i.e. heart failure vs hypertension vs arrhythmia)?

page 6, line 208: typo: 5 mg/die should be 5mg/day

    Discussion:

        - I would be careful here in stating the there is a dose-dependent correlation between carfilzomib and efficacy, despite prior studies showing this to be the case. There is the possibility patients in your study that did not receive full dose carfilzomib in the first two cycles likely had significant comorbidities that did not allow full dosing and would also contribute to poorer prognosis.  This is similar to the zoledronic acid correlation - those that did not receive likely had low creatinine clearance or poor dental health which could also correlate with poor outcomes.  Please clarify in the discussion.

        - I would be hesitant to make any conclusions regarding the significance of cytogenetics on outcomes when only 49% of patients had cytogenetic data available. 

        - page 23 line 428: typo: missing period after "proactive management"

        - page 24 line 438: typo: "worst" should be "inferior"

        - page 24 line 451: typo "combine" should be "combined"

Author Response

Response to Reviewer 1

Introduction:

p2 Line 68: I wouldn't call the prognosis of myeloma dismal anymore.  R-ISS 1 patients do not have a median survival.  I would suggest revising to say that relapse is inevitable instead.

Text has been modified according to reviewer suggestions.

p3 Line 89: Grammar error.  "To worse this scenario" should be "Worse yet, several trials have not distinguished relapsed from refractory MM,..."

Grammar error was corrected.

Methods:

Why was efficacy assessment delayed until after Cycle 2? The median time to response with KRd is 1 cycle.  Were patients not assessed (i.e. did not reach cycle 2) reported as having not reached PR or progressed or were they censored?  Please clarify in the text.  

Why were efficacy assessments performed every two months?  This would introduce a significant PFS bias, since many PFS events would be recorded a month late.  The ASPIRE study, upon which the study treatment protocol was based on, had efficacy assessments on Day 1 of each cycle.

We have clarified time points of disease assessment as follows: “Efficacy assessment was recorded  from cycle 2 and every other cycle thereafter. Efficacy assessments were performed for all patients on day 1 of each cycle from the first cycle ( page 4, line 143).

p4 line 153: you left the entire reference written out in the text, when a foot note would be more appropriate.

Reference was modified in a more appropriate way ( page 4, line 146).

Statistics: please specify if the p-value was 2-sided or not.

2-sided p-value was used and specified in methods ( page 4, line 156).

Results:

page 5: safety: 8% of the patients had Grade 3 or higher cardiovascular complications.  CV AEs are of special interest when carfilzomib is used.  Can you please clarify what the CV complications were (i.e. heart failure vs hypertension vs arrhythmia)?

We have specified grade 3 or higher cardiovascular complications according to reviewer suggestions ( page 5, line 198).

page 6, line 208: typo: 5 mg/die should be 5mg/day

Typing error was corrected.

    Discussion:

        - I would be careful here in stating the there is a dose-dependent correlation between carfilzomib and efficacy, despite prior studies showing this to be the case. There is the possibility patients in your study that did not receive full dose carfilzomib in the first two cycles likely had significant comorbidities that did not allow full dosing and would also contribute to poorer prognosis.  This is similar to the zoledronic acid correlation - those that did not receive likely had low creatinine clearance or poor dental health which could also correlate with poor outcomes.  Please clarify in the discussion.

We have clarified in the discussion the possible association between carfilzomib full dose and patients outcome: “although it should be underlined thatpatients who did not receive full dose of carfilzomib had often significant comorbidities, likely contributing to poorer outcome.( page 23, line 423).

I would be hesitant to make any conclusions regarding the significance of cytogenetics on outcomes when only 49% of patients had cytogenetic data available.

Because we described a small number of patients with evaluable cytogenetic analysis , we decided to describe our data without making any conclusions( page 24, line 470). 

- page 23 line 428: typo: missing period after "proactive management"      

- page 24 line 438: typo: "worst" should be "inferior"   

- page 24 line 451: typo "combine" should be "combined"

Typing errors were corrected (page 23, line 436; page 24, line 448; page 24, line 462).

Reviewer 2 Report

In this manuscript, authors reported the real life result of KRD therapy, derived from the analysis of 130 RRMM patients from 12 Sicilian Centers, and conclude the unfavorable efficacy of this therapy on the group with following characteristics; lower cumulative dose of carfilzomib, high risk cytogenetics, lower value of CrCL, and pre-exposure of IMids. The manuscript is well written, would provide the further insight on the practical use of this therapy. Several comments are listed as below.

     In a Table 4, the result of all 130 patients are also to be added

     Authors revealed the favorable result of 21 patients received the 18 and over cycle of KRd therapy. However, these results only identified the specific population with good sensitivity of KRd therapy, therefore, not suggested the recommendation of over-treatment by beyond 18 cycle for the all population.

     In a last paragraph of conclusion, the result of CLARION study, Carfilzomib+MP for NDMM, is recommended to be added.

Author Response

Response to Reviewer 2

Comments and Suggestions for Authors

In this manuscript, authors reported the real life result of KRD therapy, derived from the analysis of 130 RRMM patients from 12 Sicilian Centers, and conclude the unfavorable efficacy of this therapy on the group with following characteristics; lower cumulative dose of carfilzomib, high risk cytogenetics, lower value of CrCL, and pre-exposure of IMids. The manuscript is well written, would provide the further insight on the practical use of this therapy. Several comments are listed as below.

     In a Table 4, the result of all 130 patients are also to be added

In table 4 we have now added a column descibing all 130 patients 

     Authors revealed the favorable result of 21 patients received the 18 and over cycle of KRd therapy. However, these results only identified the specific population with good sensitivity of KRd therapy, therefore, not suggested the recommendation of over-treatment by beyond 18 cycle for the all population.

We have clarified that we need further studies to confirm whether  prolonged exposure to KRd is recommendedbecause we have obtained favourable results only in a small number of patients ( page 23, line 442). 

     In a last paragraph of conclusion, the result of CLARION study, Carfilzomib+MP for NDMM, is recommended to be added.

We have added in the last paragraph results of CLARION trial and inserted its reference ( page 25, line 489; reference 31). 

Round  2

Reviewer 2 Report

Agree to the acceptance. All concerns were met.